# Serum Endocan Is a Risk Factor for Aortic Stiffness in Patients Undergoing Maintenance Hemodialysis

**DOI:** 10.3390/medicina60060984

**Published:** 2024-06-14

**Authors:** Tsung-Jui Wu, Chih-Hsien Wang, Yu-Hsien Lai, Chiu-Huang Kuo, Yu-Li Lin, Bang-Gee Hsu

**Affiliations:** 1Division of Nephrology, Department of Medicine, Hualien Armed Forces General Hospital, Hualien 97144, Taiwan; ahreiwu@gmail.com; 2Institute of Medical Sciences, Tzu Chi University, Hualien 97004, Taiwan; 3Division of Nephrology, Department of Internal Medicine, Tri-Service General Hospital, National Defense Medical Center, Taipei 11490, Taiwan; 4Divisions of Nephrology, Hualien Tzu Chi Hospital, Buddhist Tzu Chi Medical Foundation, Hualien 97004, Taiwan; wangch33@gmail.com (C.-H.W.); hsienhsien@gmail.com (Y.-H.L.); hermit.kuo@gmail.com (C.-H.K.); nomo8931126@gmail.com (Y.-L.L.); 5School of Medicine, Tzu Chi University, Hualien 97004, Taiwan; 6School of Post-Baccalaureate Chinese Medicine, Tzu Chi University, Hualien 97004, Taiwan

**Keywords:** aortic stiffness, carotid–femoral pulse wave velocity, diabetes, endocan, hemodialysis, inflammation

## Abstract

*Background and Objectives*: Endocan, secreted from the activated endothelium, is a key player in inflammation, endothelial dysfunction, proliferation of vascular smooth muscle cells, and angiogenesis. We aimed to investigate the link between endocan and aortic stiffness in maintenance hemodialysis (HD) patients. *Materials and Methods*: After recruiting HD patients from a medical center, their baseline characteristics, blood sample, and anthropometry were assessed and recorded. The serum endocan level was determined using an enzyme immunoassay kit, and carotid–femoral pulse wave velocity (cfPWV) measurement was used to evaluate aortic stiffness. *Results*: A total of 122 HD patients were enrolled. Aortic stiffness was diagnosed in 53 patients (43.4%), who were found to be older (*p* = 0.007) and have a higher prevalence of diabetes (*p* < 0.001) and hypertension (*p* = 0.030), higher systolic blood pressure (*p* = 0.011), and higher endocan levels (*p* < 0.001), when compared with their counterparts. On the multivariate logistic regression model, the development of aortic stiffness in patients on chronic HD was found to be associated with endocan [odds ratio (OR): 1.566, 95% confidence interval (CI): 1.224–2.002, *p* < 0.001], age (OR: 1.040, 95% CI: 1.001–1.080, *p* = 0.045), and diabetes (OR: 4.067, 95% CI: 1.532–10.798, *p* = 0.005), after proper adjustment for confounders (adopting diabetes, hypertension, age, systolic blood pressure, and endocan). The area under the receiver operating characteristic curve was 0.713 (95% CI: 0.620–0.806, *p* < 0.001) for predicting aortic stiffness by the serum endocan level, at an optimal cutoff value of 2.68 ng/mL (64.15% sensitivity, 69.57% specificity). Upon multivariate linear regression analysis, logarithmically transformed endocan was proven as an independent predictor of cfPWV (β = 0.405, adjusted R^2^ change = 0.152; *p* < 0.001). *Conclusions*: The serum endocan level positively correlated with cfPWV and was an independent predictor of aortic stiffness in chronic HD patients.

## 1. Introduction

In Taiwan, although the incidence of end-stage renal disease (ESRD) on maintenance dialysis has numerically decreased for three consecutive years, it remains the highest worldwide, with a prevalence of 3839 per million population in 2021 [1]. Similar to the situation in the United States, cardiovascular disease (CVD) remains the major comorbidity at the end life of a patient on dialysis [2]. Currently, consensus on the overall benefit of intensive blood pressure control has been demonstrated in many randomized clinical trials, including the Modification of Diet in Renal Disease, African American Study of Kidney Disease and Hypertension, Action to Control Cardiovascular Risk in Diabetes, Systolic Blood Pressure Intervention Trial, and Strategy of Blood Pressure Intervention in the Elderly Hypertensive Patients [3,4,5,6]. However, in a meta-analysis of the aforementioned studies, the beneficial effect was found to be not applicable to patients on maintenance dialysis [7]. In other words, there is a clear unmet need in the clinical care of patients with ESRD.

Aortic stiffness represents increased cardiac workload and is a proven mechanical biomarker for cardiovascular morbidity and mortality in patients with chronic kidney disease (CKD), with or without dialysis [8,9]. The severity of aortic stiffness could be noninvasively quantified by the velocity of pulse pressure transmission along the arterial tree [10]. Although pulse wave velocity is well recognized for its predictive power, limited availability, various devices, and the requirement of a specialist to perform the measurement were the factors that prevented its widespread adoption in clinical practice [11].

Many clinical studies focused on the correlation between the severity of aortic stiffness in patients with ESRD and various serum biomarkers, such as uremic toxin and inflammatory/atherosclerosis markers [12,13]. In the event of vascular inflammation, adipocytes produce both proinflammatory molecules, such as tumor necrosis factor alpha (TNFα), and anti-inflammatory molecules, such as adiponectin, to interfere with the function of monocytes/macrophages and the production of endocan, which was previously known as endothelial cell-specific molecule-1 (ESM1) [14,15,16]. In 1996, endocan was first identified as a 20 kDa proteoglycan with a limited expression pattern on endothelial cells of the lungs, kidneys, and umbilical cord [17]. Wellner et al. demonstrated that ESM1 formation in adipocytes is induced by protein kinase C, suggesting its central role in regulating inflammatory processes [18]. The diagnostic and prognostic role of endocan has also been extensively investigated and evidenced in the various events of lung inflammation [19]. Muhammad et al. demonstrated that, clinically, even low-grade inflammation was associated with arterial stiffness in a large, prospectively population-based cohort in Sweden [20]. Aortic PWV was also demonstrated to be positively associated with the inflammatory disease duration and white blood cell count in a systemic meta-analysis study [21]. Khalaji et al. further showed that a relatively high endocan level was associated with CVD risk in patients with impaired renal function [22]. This present study investigated the association of the serum endocan level with a mechanical biomarker for aortic stiffness, as measured by the carotid–femoral pulse wave velocity (cfPWV), in patients on maintenance hemodialysis (HD).

## 2. Materials and Methods

### 2.1. Participants and Study Design

After the study protocol was approved by the research ethics committee of Hualien Tzu Chi Hospital, Buddhist Tzu Chi Medical Foundation (IRB109-121-C), patients > 20 years old and on thrice weekly maintenance 4 h standard HD for >6 months at a medical center in eastern Taiwan from 1 July 2021 to 30 September 2021 were screened for this study. The exclusion criteria were malignancy, cerebrovascular accident, acute infection, limb amputation, heart failure, liver cirrhosis, chronic obstructive lung disease, and immobilization. A total of 122 patients on HD were included and provided signed informed consent. Figure 1 depicts the flow chart of this study. The medical records were reviewed for the etiology of renal failure. The diagnoses of diabetes mellitus (DM) and hypertension were confirmed based on the documentation or prescription of antidiabetic or antihypertensive agents. Furthermore, the use of renin–angiotensin system blockers, beta blockers, calcium channel blockers, statins, and fibrates was specifically recorded. This study was conducted following the ethical guidelines of the Declaration of Helsinki.

### 2.2. Anthropometric Analyses and Biochemical Investigations

Blood pressure was taken before the HD sessions after a 10 min seated rest. The body mass index was calculated by dividing the body weight (in kg) measured after HD by the body height (in m^2^). Upon HD initiation, approximately 5 mL of blood was drawn from each participant. Once the hemoglobin concentration was determined (Sysmex SP-1000i, Sysmex American, Mundelein, IL, USA), the rest of the blood sample was immediately processed by centrifugation at 3000× *g* for 10 min. For biochemical analyses, the levels of blood urea nitrogen, creatinine, total cholesterol, triglyceride, glucose, albumin, calcium, and inorganic phosphorus were measured by an autoanalyzer (SiemensAdvia 1800, Siemens Healthcare GmbH, Henkestr, Germany). All HD sessions were performed using high-flux polysulfone and non-reused dialyzers (FX class dialyzer; Fresenius Medical Care, Bad Homburg, Germany). The adequacy of HD was determined by urea clearance (Kt/V), using the single pool urea kinetic modeling. The serum levels of endocan (Aviscera Bioscience, Inc., Santa Clara, CA, USA) and intact parathyroid hormone (iPTH) (Abcam, Cambridge, MA, USA) were measured using commercially available enzyme-linked immunosorbent assay kits [12,13].

### 2.3. Assessments of Aortic Stiffness by Applanation Topometry

After asking the participant to relax in a supine position for at least 10 min, the cfPWV was measured using transcutaneous SphygmoCor probe (SphygmoCor system, AtCor Medical, Sydney, Australia), which simultaneously recorded the pulse waveform in the target artery area and electrocardiogram (ECG) [12,13]. After obtaining an accurate and reproducible pulse waveform, we recorded at least 12 s of signal or 10 consecutive cardiac cycles. The arterial path length required for pulse wave velocity calculation was obtained by subtracting the carotid length from the distance between the suprasternal notch and the femoral recording site. Each set of pulse waveforms and ECG recordings were entered into the SphygmoCor software 1.30 (AtCor Medical) for an integrated calculation of the average time difference between the ECG R waves and the pulse waveforms of the two recording sites on a beat-to-beat basis. Thereafter, the cfPWV was automatically generated by the software as the arterial path length divided by the travel time of the pulse wave between two sites. Measurements with <10% standard deviations were included for analysis, and other quality indices were applied to ensure data uniformity. The participant was classified in the aortic stiffness group if the cfPWV was >10 m/s; this represented an increased cardiovascular risk, according to the consensus of the 2018 European Society of Cardiology clinical practice guidelines on arterial hypertension [23].

### 2.4. Statistical Analysis

All collected data were first examined for normality using Kolmogorov–Smirnov test. Normally distributed data were expressed as mean ± standard deviation and were compared between the two groups using two-tailed independent Student’s *t*-test. Data with nonnormal distribution, such as HD duration, cfPWV, triglyceride, glucose, iPTH, and endocan, were expressed as median (interquartile range) and were compared by the Mann–Whitney U test. To conform to normality, skewness was reduced by common logarithmical transformation. The categorical variables were presented as number (%) and analyzed by the χ^2^ test. Variables that were significantly associated with aortic stiffness were examined in the multivariate logistic regression model. The simple linear regression model evaluated the clinical variables that correlated with the cfPWV. Those that were significantly associated with the cfPWV were tested for independence using a multivariate forward stepwise regression model. A receiver operating characteristic (ROC) curve was used to examine the diagnostic performance of serum endocan level in predicting aortic stiffness, and the area under the curve (AUC) was calculated to determine accuracy. Data analyses were performed using the statistical software SPSS for Windows (version 19.0; SPSS Inc., Chicago, IL, USA). Statistical significance was set at a *p* value of <0.05.

## 3. Results

### 3.1. Baseline Characteristics

The demographic and biochemical characteristics of the participants are summarized in Table 1. Patients with increased cfPWV were assigned to the aortic stiffness group, and those with normal cfPWV were assigned to the control group. The 122 participants, in total, comprised 53 (43.4%) in the aortic stiffness group and 69 (56.6%) in the control group. There were no differences between groups in terms of sex, HD duration, anthropometric data, general blood biochemistries, HD adequacy, and use of antihypertensive or lipid-lowering agents. Compared with the control group, the aortic stiffness group was significantly older (*p* = 0.007) and had significantly higher systolic blood pressure (SBP, *p* = 0.011), serum endocan level (*p* < 0.001), and prevalence of diabetes mellitus (DM, *p* < 0.001) and hypertension (*p* = 0.030).

### 3.2. Serum Endocan Level and Development of Aortic Stiffness

Table 2 shows the relationships between the clinical variables and aortic stiffness. After proper adjustments for confounding factors (including DM, hypertension, age, SBP, and endocan), the development of aortic stiffness was associated with the endocan level [odds ratio (OR): 1.566; 95% confidence interval (CI): 1.224–2.002, *p* < 0.001], age (OR: 1.040, 95% CI: 1.001–1.080, *p* = 0.045), and DM (OR: 4.067, 95% CI: 1.532–10.798, *p* = 0.005). The relationships between cfPWV and each risk factor were also demonstrated in scattered bar plots (for DM, Appendix A) or regression plots (for age, SBP, and log-endocan, Appendix A).

In the ROC curve, we demonstrated the diagnostic performance of the serum endocan level for aortic stiffness, with a decent AUC of 0.713 (95% CI: 0.620–0.806, *p* < 0.001) at an optimal cutoff value of 2.68 ng/mL, with 64.15% sensitivity, 69.57% specificity, 61.82% positive predictive value, and 71.64% negative predictive value (Figure 2).

### 3.3. Correlations between cfPWV and Clinical Variables

Table 3 shows the correlations between the cfPWV and clinical variables. The cfPWV was positively correlated with age (*r* = 0.198, *p* = 0.029); systolic blood pressure (*r* = 0.264, *p* = 0.003); logarithmically transformed endocan (log-endocan, *r* = 0.399, *p* < 0.001); and the presence of DM (*r* = 0.376, *p* < 0.001). Furthermore, the stepwise multivariate regression analysis showed that the log-endocan level (β = 0.405, adjusted R^2^ change = 0.152, *p* < 0.001) and DM status (β = 0.381, adjusted R^2^ change = 0.141, *p* < 0.001) were the independent predictors of the cfPWV in patients on HD.

## 4. Discussion

In this study, we demonstrated that among patients on maintenance HD, those who had aortic stiffness were relatively older and had a relatively high prevalence of DM and hypertension, systolic blood pressure, and endocan levels. The odds ratio for aortic stiffness increased by 56.6% with each 1 ng/mL increase in the serum endocan level. Moreover, the serum endocan level exerted a decent predictive performance for aortic stiffness and was independently positively correlated with the cfPWV.

In patients on HD, sodium accumulation from the diet and the lack of self-regulating body fluid homeostasis result in hypervolemia. Consequently, stiffness and remodeling of the left ventricle develop and progress to heart failure [24,25,26]. In addition to several factors, such as increased systemic sympathetic tone and vascular resistance, inflammation, reduced renal toxin excretion, maladaptation of metabolic/hormonal processes [27,28], arterial stiffness appeared to contribute to the accelerated vascular aging in uremia. Overactivation of the renin–angiotensin–aldosterone system results from the rapid shift of salt and extracellular volume, which is also evident during increased renin activity and aldosterone levels after an HD session [29,30]. On the other hand, endothelial dysfunction can also be present in apparent CKD, acute kidney injury, and even local exposure to uremic toxins in healthy volunteers [31,32]. In an in vitro experiment on subcutaneous arterial resistance in patients with uremia, Morris et al. concluded that endothelial dysfunction was secondary to impaired nitric oxide (NO) production in uremic vessels [33]. In an animal study, Vaziri et al. successfully demonstrated that NO inactivation resulted from the overproduction of reactive oxygen species in a CKD rat and can be ameliorated by antioxidant treatment with high-dose vitamin E [34].

Endocan is a key proteoglycan that integrates with glycoproteins to form the luminal endothelial surface layer, which forms the blood vessel barrier and senses the mechanical transduction of laminar blood flow shear stress to regulate the NO-mediated signaling pathways that play a central role in maintaining vascular homeostasis [35]. Glycocalyx injury, which results in the release of its components into the bloodstream, can be regarded as an early event in endothelial dysfunction and can be quantified by the endocan level [36]. Circulating endocan at the sites of inflammation regulates major processes, such as cell adhesion, and is even involved in tumor progression. Moreover, accumulating experimental evidence suggests that inflammatory cytokines, such as TNFα, and growth factors with proangiogenic potential (e.g., vascular endothelial growth factor, fibroblast growth factor 2, and hepatocyte growth factor/scatter factor hepatocyte growth factor/scatter factor) were strongly associated with the upregulated expression, synthesis, and/or secretion of endocan from the human endothelium [37].

Several clinical studies have addressed the serum endocan level in patients with various severities of renal impairment. Oka et al. found correlations among the serum endocan level, serum TNFα, and the decline in residual urine volume in patients on peritoneal dialysis [38]. Samouilidou et al. showed that increased endocan levels in patients with CKD might be associated with a decreased level of the antiatherogenic protein paraoxonase 1 [39]. In patients with stage 5 CKD without dialysis, endocan seemed to be associated with the circadian heart rate variability [40]. Furthermore, there have been clinical studies that revealed the association of endocan with worse cardiovascular outcomes in patients on peritoneal dialysis [41] and with cardiovascular events and all-cause mortality in patients on HD [42,43]. In a meta-analysis, the endocan level was found to be higher in patients with CKD than in healthy controls, despite the high heterogeneity, and was also demonstrated as a negative prognostic factor in patients on HD, peritoneal dialysis, and even after renal transplantation [22]. The results of our study supported the abovementioned findings and contributed evidence for the clinical use of endocan for CVD surveillance in patients on maintenance HD.

There were several limitations in our study. The nature of a cross-sectional study from a single center might hamper the generalizability of our findings. Endocan could be affected by various conditions, such as dialysis treatment modality or medications [44,45]. Although we excluded patients with acute stress, such as acute infection, cerebrovascular accident, heart failure, liver cirrhosis, chronic obstructive lung disease, malignancy, or chronic deconditioning, such as limb amputation and immobilization, patients in the prodromal stage of this devastating disease could have been easily overlooked, especially those with relatively immune-compromised status secondary to uremia. Diabetes is strongly associated with arterial stiffness, and vice versa [46]. We analyzed only the impact of the presence of diabetes, without further investigating the disease duration. Future research should tackle these shortcomings by including more participants under different dialysis treatment modalities for further subgroup analyses.

## 5. Conclusions

This emerging endothelial dysfunction biomarker, the serum endocan level, was demonstrated to positively correlate with the surrogate marker of aortic stiffness in patients on maintenance HD, not only in a dose-dependent manner but also with decent predictive power.

## Figures and Tables

**Figure 1 medicina-60-00984-f001:**
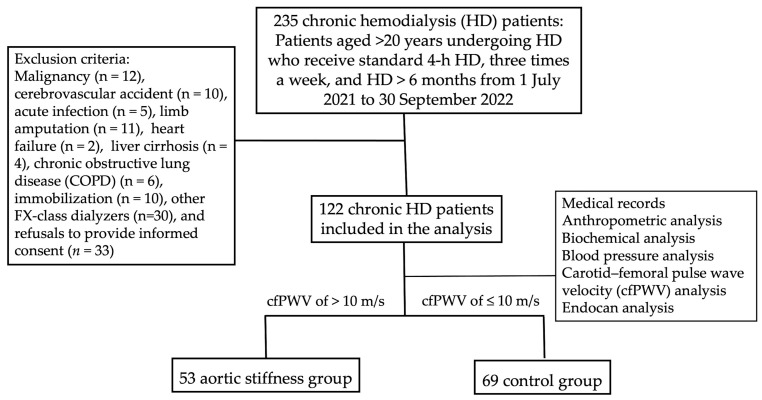
Flow chart of this study.

**Figure 2 medicina-60-00984-f002:**
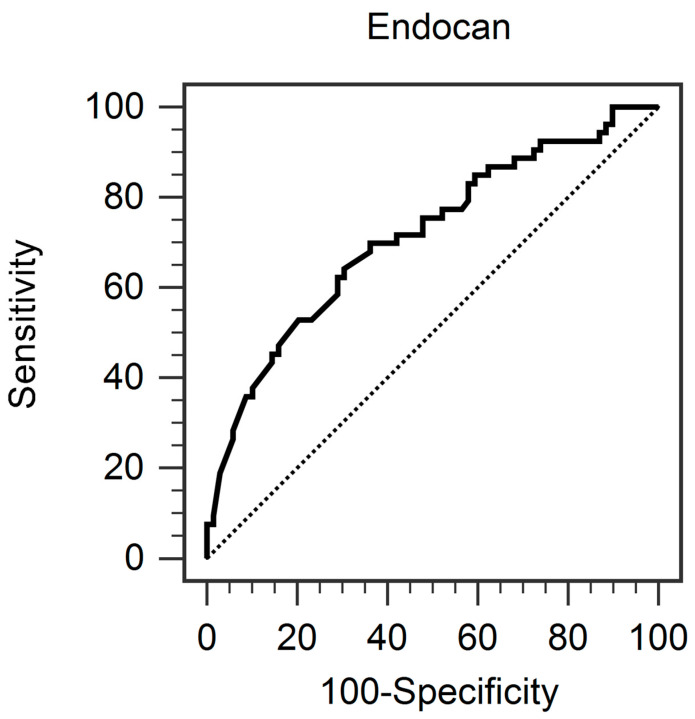
The receiver operating characteristic (ROC) curve of serum endocan level on diagnosing aortic stiffness in the 122 hemodialysis patients. The area under curve (AUC) was 0.713 (95% CI: 0.620–0.806; *p* < 0.001) at an optimal cutoff value of 2.68 ng/mL, with 64.15% sensitivity, 69.57% specificity, 61.82% positive predictive value, and 71.64% negative predictive value.

**Table 1 medicina-60-00984-t001:** Clinical variables of the 122 hemodialysis patients with or without aortic stiffness.

Characteristics	All Patients (*n* = 122)	Control Group (*n* = 69)	Aortic Stiffness Group (*n* = 53)	*p* Value
Age (years)	63.28 ± 12.26	60.68 ± 13.10	66.66 ± 10.24	0.007 *
Female, *n* (%)	61 (50.0)	37 (53.6)	24 (45.3)	0.361
HD duration (months)	58.26 (25.53–129.60)	80.76 (21.66–144.00)	53.40 (26.70–85.44)	0.268
Height (cm)	159.97 ± 8.59	159.51 ± 9.04	160.57 ± 8.01	0.502
Pre-HD body weight (kg)	63.60 ± 15.10	62.84 ± 15.48	64.59 ± 14.67	0.528
Post-HD body weight (kg)	61.41 ± 14.72	60.70 ± 15.04	62.33 ± 14.37	0.545
Body mass index (kg/m^2^)	24.89 ± 4.91	24.71 ± 5.24	25.12 ± 4.50	0.652
Carotid-femoral PWV (m/s)	9.25 (7.50–12.45)	7.70 (7.00–8.80)	12.90 (11.75–14.90)	<0.001 *
Systolic blood pressure (mmHg)	141.59 ± 24.60	136.67 ± 23.52	148.00 ± 24.71	0.011 *
Diastolic blood pressure (mmHg)	76.01 ± 15.03	75.84 ± 14.64	76.36 ± 15.66	0.851
Hemoglobin (g/dL)	10.47 ± 1.17	10.36 ± 1.26	10.62 ± 1.05	0.229
Albumin (g/dL)	4.16 ± 0.47	4.16 ± 0.47	4.16 ± 0.47	0.995
Total cholesterol (mg/dL)	145.58 ± 34.12	149.20 ± 37.95	140.87 ± 28.03	0.182
Triglyceride (mg/dL)	113.00 (86.75–178.00)	109.00 (88.00–193.00)	118.00 (85.00–169.00)	0.916
Glucose (mg/dL)	131.00 (108.50–164.00)	129.00 (103.50–151.00)	132.00 (112.50–179.00)	0.187
Blood urea nitrogen (mg/dL)	60.48 ± 14.77	59.74 ± 13.06	61.45 ± 16.82	0.527
Creatinine (mg/dL)	9.32 ± 2.06	9.44 ± 2.04	9.16 ± 2.09	0.455
Total calcium (mg/dL)	9.03 ± 0.78	8.95 ± 0.75	9.14 ± 0.80	0.185
Phosphorus (mg/dL)	4.65 ± 1.32	4.61 ± 1.33	4.70 ± 1.31	0.718
iPTH (pg/mL)	194.95 (58.50–363.05)	239.30 (94.70–366.20)	141.10 (55.55–357.60)	0.287
Endocan (mg/L)	26.63 (14.59–44.80)	18.35 (10.86–30.42)	32.17 (17.89–53.31)	<0.001 *
Urea reduction rate	0.74 ± 0.04	0.74 ± 0.04	0.73 ± 0.04	0.397
Kt/V (Gotch)	1.35 ± 0.17	1.36 ± 0.18	1.33 ± 0.15	0.332
Diabetes mellitus, *n* (%)	52 (42.6)	19 (27.5)	33 (62.3)	<0.001 *
Hypertension, *n* (%)	60 (49.2)	28 (40.6)	32 (60.4)	0.030 *
RAS blocker, *n* (%)	38 (31.1)	19 (27.5)	19 (35.8)	0.236
Beta-blocker, *n* (%)	40 (32.8)	21 (30.4)	19 (35.8)	0.528
Calcium channel blocker, *n* (%)	48 (39.3)	28 (40.6)	20 (37.7)	0.750
Statin, *n* (%)	26 (21.3)	13 (18.8)	13 (24.5)	0.447
Fibrate, *n* (%)	27 (22.1)	18 (26.1)	9 (17.0)	0.230

Values of continuous variables were expressed as mean ± standard deviation, analyzed by the *t*-test; non-normally distributed variables were shown as median (interquartile ranges), analyzed by Mann–Whitney U test; categorical values were presented as number (%), analyzed by the χ^2^ test. HD, hemodialysis; Kt/V, fractional clearance index for urea. RAS, renin–angiotensin system. * *p* < 0.05 was considered statistically significant.

**Table 2 medicina-60-00984-t002:** Multivariate logistic regression analysis of the clinical variables related to aortic stiffness among the 122 hemodialysis patients.

Variables	Odds Ratio	95% Confidence Interval	*p* Value
Endocan, 1 ng/mL	1.566	1.224–2.002	<0.001 *
Age, 1 year	1.040	1.001–1.080	0.045 *
Diabetes mellitus, present	4.067	1.532–10.798	0.005 *
Systolic blood pressure, 1 mmHg	1.013	0.985–1.041	0.368
Hypertension, present	1.309	0.406–4.218	0.652

Analysis of data was done using the multivariate logistic regression analysis (adopted factors: diabetes mellitus, hypertension, age, systolic blood pressure, and endocan). * *p* < 0.05 was considered statistically significant.

**Table 3 medicina-60-00984-t003:** Correlations between carotid–femoral pulse wave velocity values and clinical variables.

Variables	Carotid-Femoral Pulse Wave Velocity				
	Uni-Variable Regression		Multi-Variable Regression		
	*r*	*p* Value	Beta	Adjusted R^2^ Change	*p* Value
Female	−0.141	0.120	–	–	–
Diabetes mellitus	0.376	<0.001 *	0.381	0.141	<0.001 *
Hypertension	0.170	0.061	–	–	–
Age (years)	0.198	0.029 *	–	–	–
Log-HD duration (months)	−0.135	0.139	–	–	–
Height (cm)	0.112	0.221	–	–	–
Pre-HD body weight (kg)	0.101	0.269	–	–	–
Post-HD body weight (kg)	0.097	0.287	–	–	–
Body mass index (Kg/m^2^)	0.067	0.462	–	–	–
Systolic blood pressure (mmHg)	0.264	0.003 *	–	–	–
Diastolic blood pressure (mmHg)	0.079	0.387	–	–	–
Hemoglobin (g/dL)	0.145	0.111	–	–	–
Albumin (g/dL)	0.066	0.473	–	–	–
Total cholesterol (mg/dL)	−0.077	0.399	–	–	–
Log-Triglyceride (mg/dL)	0.056	0.543	–	–	–
Log-Glucose (mg/dL)	0.165	0.070	–	–	–
Blood urea nitrogen (mg/dL)	0.095	0.296	–	–	–
Creatinine (mg/dL)	−0.006	0.947	–	–	–
Total calcium (mg/dL)	0.070	0.440	–	–	–
Phosphorus (mg/dL)	0.070	0.440	–	–	–
Log-iPTH (pg/mL)	−0.103	0.259	–	–	–
Log-Endocan (ng/mL)	0.399	<0.001 *	0.405	0.152	<0.001 *
Urea reduction rate	−0.073	0.425	–	–	–
Kt/V (Gotch)	−0.081	0.377	–	–	–

Data including HD duration, triglyceride, glucose, iPTH, and endocan levels showed skewed distribution and were log-transformed before analysis. Analysis was performed using univariate linear regression and multivariate stepwise linear regression analyses (adopted factors: diabetes mellitus, age, systolic blood pressure, and endocan). HD, hemodialysis; iPTH, intact parathyroid hormone; Kt/V, fractional clearance index for urea. * *p* < 0.05 was considered statistically significant.

## Data Availability

The dataset used, analyzed, or generated in this research is available from the corresponding author, B.-G.H., upon reasonable request.

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
