# Peer review of "Serum Endocan Is a Risk Factor for Aortic Stiffness in Patients Undergoing Maintenance Hemodialysis"

_medicina, 2024, doi:10.3390/medicina60060984_

Round 1

Reviewer 1 Report

Comments and Suggestions for Authors

Thank you for your article; you have done excellent work.

However, below, I have some minor suggestions:

  1. I suggest the authors add some words in the title and make it more scientific: Serum endocan is a risk factor for aortic stiffness in patients undergoing maintenance hemodialysis.
  2. In the results section, you found significant correlations between cfPWV and diabetes mellitus, age, systolic blood pressure, and log-endocan level. Authors should consider providing partial regression plots for the significant correlations mentioned above. 

Author Response

Minor issues:

  1. I suggest the authors add some words in the title and make it more scientific: Serum endocan is a risk factor for aortic stiffness in patients undergoing maintenance hemodialysis.

Response

Thank you very much for this insightful comment. The title had been re-written accordingly and the changes were highlighted in red font. (Page 1, Line 2)

  1. In the results section, you found significant correlations between cfPWV and diabetes mellitus, age, systolic blood pressure, and log-endocan level. Authors should consider providing partial regression plots for the significant correlations mentioned above.

Response

Thank you very much for these important comments. The pertinent scattered-bar plot or regression plots were included as supplementary figure 1, 2, also mentioned in the Results section (Page 5, Line 175-177 )

Reviewer 2 Report

Comments and Suggestions for Authors

There are no major issues in the paper titled “Serum endocan as a risk factor for aortic stiffness in patients on maintenance hemodialysis”.

Minor issues:

Please rephrase the following statement in abstract in sake of clarity. “Aortic stiffness was diagnosed in 53 patients (43.4%), who were found to be older (p = 0.007) and have higher prevalence of diabetes (p < 0.001) and hypertension (p = 0.030), systolic blood pressure (p = 0.011), and endocan levels (p < 0.001), when compared with those who had no aortic stiffness.”.

Authors stated that “On the multivariate logistic regression model, development of aortic stiffness in patients on chronic HD was found to be associated with endocan [odds ratio (OR): 1.566, 95% confidence interval (CI): 1.224–2.002, p < 0.001]; age (OR: 1.040, 95% CI: 1.001–1.080, p = 0.045); and diabetes (OR: 4.067, 95% CI: 1.532–10.798, p = 0.005), after proper adjustment for confounding factors.”. Please mention confounding factors in abstract and in the text.

Authors stated that “The area under the receiver operating characteristic curve was 0.713 (95% CI: 0.620–0.806, p < 0.001) for predicting aortic stiffness by the serum endocan level.”. Please express the sensitivity and specifity properly.

Inflammation should be added as another keyword since endocan is associated with inflammatory conditions and aortic stiffness has been suggested to be associated with high burden of inflammation.

Inflammatory association of endocan (Crit Care. 2018 Oct 26;22(1):280. doi: 10.1186/s13054-018-2222-7) must be emphasized in introduction along with emphasizing inflammatory features of aortic stiffness (J Am Heart Assoc. 2017 Oct 10;6(10):e007003. doi: 10.1161/JAHA.117.007003).

Methodology is near perfect and results were expressed fairly. However, significant p values could be given as bold characters.

The ROC curve for endocan shown in figure 2 should be prepared and presented in a clearer (finer line) and more understandable manner.

Discussion is fair. Limitations are justified. Maybe authors should acknowledge single center nature of the work as another limitation.

References 12,13 and 39 looks like sel-citations.

Author Response

Minor points:

  1. Please rephrase the following statement in abstract in sake of clarity. “Aortic stiffness was diagnosed in 53 patients (43.4%), who were found to be older (p = 0.007) and have higher prevalence of diabetes (p < 0.001) and hypertension (p = 0.030), systolic blood pressure (p = 0.011), and endocan levels (p < 0.001), when compared with those who had no aortic stiffness.”

Response

Thank you very much for this kind reminder. The abstract had been re-written accordingly and the changes were high-lighted in red font. (Page 1, Line 25-26)

  1. Authors stated that “On the multivariate logistic regression model, development of aortic stiffness in patients on chronic HD was found to be associated with endocan [odds ratio (OR): 1.566, 95% confidence interval (CI): 1.224–2.002, p < 0.001]; age (OR: 1.040, 95% CI: 1.001–1.080, p = 0.045); and diabetes (OR: 4.067, 95% CI: 1.532–10.798, p = 0.005), after proper adjustment for confounding factors.”. Please mention confounding factors in abstract and in the text.

Response

Thank you very much for this kind reminder. The statement regarding multivariate logistic regression modelhad been re-written to include those adopted confounding factors and the changes were high-lighted in red font. (Page 1, Line 29-30; Page 5, Line 172-173)

  1. Authors stated that “The area under the receiver operating characteristic curve was 0.713 (95% CI: 0.620–0.806, p < 0.001) for predicting aortic stiffness by the serum endocan level.”. Please express the sensitivity and specificity properly.

Response

Thank you very much for this kind reminder. The statement regarding ROC curve analysis had been re-written to include an optimal cutoff value with sensitivity and specificity, and the changes were high-lighted in red font. (Page 1, Line 32-33; Page 5, Line 179-181)

  1. Inflammation should be added as another keyword since endocan is associated with inflammatory conditions and aortic stiffness has been suggested to be associated with high burden of inflammation.

Response

We thank the reviewer for this insightful comment. The keywords had been re-written to include inflammation and high-lightened in red font. (Page 1, Line 38)

  1. Inflammatory association of endocan (Crit Care. 2018 Oct 26;22(1):280. doi: 10.1186/s13054-018-2222-7) must be emphasized in introduction along with emphasizing inflammatory features of aortic stiffness (J Am Heart Assoc. 2017 Oct 10;6(10):e007003. doi: 10.1161/JAHA.117.007003).

Response

We thank the reviewer for this insightful comment. The Introduction section had been re-written to include the possible associations between inflammation, endocan, and aortic stiffness, also high-lighted in red font. (Page 2, Line 72-78)

  1. Methodology is near perfect and results were expressed fairly. However, significant p values could be given as bold characters.

Response

We thank the reviewer for this kind reminder. The significant p values had been presented in bold font in all following paragraphs.

  1. The ROC curve for endocan shown in figure 2 should be prepared and presented in a clearer (finer line) and more understandable manner.

Response

We thank the reviewer for this kind reminder. The ROC curve had been reproduced in a finer line and the caption had been rephrased to describe in detail (Page 6, Figure 2).

  1. Discussion is fair. Limitations are justified. Maybe authors should acknowledge single center nature of the work as another limitation.

Response

We thank the reviewer for this kind reminder. The subsection of Limitation had been re-written to include the nature of a single-center study. (Page 8, Line 259-260 )

  1. References 12,13 and 39 looks like self-citations.

Response

Thank you very much for this critical comment. Our team collaborate with other groups from medical centers across Taiwan, sharing the same interest of research — vascular calcification and its clinical applications — which we have been focusing on for many years. With the advance of technology, more biomarkers from transcriptomic and proteomic data are found linked to this matter. To evaluate the diagnostic/prognostic values of these biomarkers, clinical studies to examine each and every one are necessary. Therefore, you may find similar topics that had been published by our group or our collaborators, but all different in terms of biomarker-of-interest or patient population. We would like to thank the reviewer for the opportunity to explain why these citations are crucial and to make improvement of our work.